# Invigorating Health Strategy in an Integrated Design Process

**Stahel Serano Bibang Bi Obam Assoumou [1],\*, Li Zhu [1] and Christopher Khayeka-Wandabwa [2]**

1   School of Architecture, Tianjin University, Tianjin 300072, China; zhuli1977@tju.edu.cn
2   School of Pharmaceutical Science and Technology, Healthy Sciences Platform, Tianjin University, Tianjin 300072, China; khayekachris@yahoo.com
\*   Correspondence: bibangserano@gmail.com or bibang@tju.edu.cn

**Abstract:** Healthy buildings are gaining crucial significance in construction and one health setting for promoting occupants' health. However, the traditional design process for healthy buildings presents limitations with no specific guidelines. In contrast, the integrated design process (IDP) has proven to be effective in realizing high-performance buildings. However, the IDP shortfall of not having robust health strategy (HS) capability is a concern of interest. Thus, we posit further advancement of IDP in the context of incorporating HS in the prevailing IDP guidelines with a sequential iterative procedure. Moreover, a conceptual framework aimed at invigorating the implementation of HS in all IDP stages is proposed. The strategies within IDP that would reinforce achieving healthy building by addressing building process implementation are highlighted. The (IDP + HS) iterative framework herein advanced is intended to aid neophyte and experienced building professionals to reflect about the process of achieving healthy building while optimizing IDP for one's health invigoration in construction industry.

**Keywords:** integrated design process; traditional design process; healthy building; indoor environment quality





## 1. Introduction

The fast-advancing technology in every facet of life and urbanization witnessed in the 21st century has brought forth an increasing demand for healthy buildings for the work environment and housing as a critical component of human daily life [1]. In the United Nations 17 Sustainable Development Goals (SDGs), goal 3, good health and well-being; and goal 8, decent work and economic growth [2] are the two scopes where healthy building expertise and technological advances resonate well. Today, people spend the majority of their time indoors, as a result bringing a sharp focus on building conditions concerning occupant well-being, comfort, and health status [3]. Indoor environmental quality (IEQ) can be a leading source of illness and have a detrimental influence on occupant comfort, health, and well-being, particularly in the office and home environments where people spend more time [4].

Advances in sustainable design and construction and scientific and medical technology have resulted in a stronger demand for a healthy living and working environments than ever before [5], emphasizing the need for a healthy built environment as part of one's overall health. A healthy building environment includes acceptable indoor air quality, minimal noise and vibrations, adequate lighting, security, acoustics, and comfortable temperature and humidity, as well as safety and ergonomic design factors. The built environment is defined as any physical buildings developed and designed by humans, including places where people work, live, socialize, and play [6]. It also includes material determinants of health, all of which shape diverse crucial, subtle elements of one's health like the environmental conditions on which good health is dependent. There is a growing acknowledgment that risk factors and/or causes of health complications including heart disease, cancer, cerebrovascular disease, respiratory diseases, and injuries may be worsened

by elements within the built environment that contribute to sedentary lifestyles and harmful environments [7]. To solve the issues, the stages of design, construction, and operation have been thoroughly examined.

There have been various attempts in recent years to describe the features and methods for developing healthy buildings. In this regard, the emphasis on healthy building design has been on implementing integrated and comprehensive processes across the planning, design, and construction phases of a building's lifecycle to create a holistic perspective and teamwork [8]. The advancements have been driven by leveraging changing paradigms from the traditional design process (TDP) to the integrated design process (IDP) with a sharp focus on healthy building standards [9–11]. Thus, we posit further advancement of IDP in the context of incorporating HS in the prevailing IDP healthy building construction guidelines with a sequential iterative procedure. Moreover, a conceptual framework of IDP + HS aimed at invigorating the implementation of HS in all IDP workflow stages is underscored. The strategies within IDP that lead to achieving healthy construction by addressing both indoor and general environmental quality are also emphasized.

The outcome of this study can help professionals to successfully implement healthy construction strategies and technologies throughout the entire design process and development that can eventually achieve the goals of health in the building industry. In addition, the proposed framework can also serve as a useful education and training tool for healthy construction that will help construction professionals to learn about their professional responsibilities and the resulting benefits for the construction industry as a whole.

## 2. Design Process and Paradigm Shift to Healthy Building

Despite the well-accepted advances from the traditional design process (TDP) to the integrated design process (IDP), the concept of a healthy building remains contentious, with no consistent guidelines. The traditional design process (TDP) used to develop buildings over the years may have contributed to poor health conditions, due in part to the separation of responsibility among professions involved in the design and construction process. The design, bidding, and construction of structures, as well as their role in the context of sick building syndrome, have been demonstrated [9]. It was found that the standard design process method and its variants in various regions of the world were highly unlikely to contribute to healthy construction. This was due to the fact that, in TDP, the design and construction teams are not always interactive. They have different arrangements, have different responsibilities, and do not participate in the project throughout. Even when high-performance systems are built, they may not perform as expected due to poor coordination during the design and construction processes [10].

Contrary to TDP, numerous studies have stressed the importance of an integrated design process (IDP) with a framework that can help professionals implement IDP in a step-by-step teamwork procedure. IDP is a holistic approach to construction since it provides the roles and responsibilities of major team members in stepwise guidelines. The mechanism provides the means to apply the design strategies towards sustainability [8]. However, the shortfall of IDP not having a robust health strategy (HS) capability is a concern of interest. Thus, it is agreeable that healthy building design incorporates a health strategy, must adopt the integrated and comprehensive processes outlined under the IDP approach that is implemented during the design, and should design the construction stages of a building's lifecycle to create a holistic perspective of expertise cooperation and build environment [11].

### 2.1. Healthy Building Concept and Trends

Healthy buildings are considered to be high-performance buildings that promote occupant comfort, health, safety, and environmental responsibility while also meeting the physiological, psychological, and social needs of occupants [5]. The notion of a healthy building was identified by the World Health Organization (WHO, Geneva, Switzerland) around the 1980s as free of harmful materials and able to promote the health and com-

fort of the occupants while meeting social needs [12]. The healthy building approach was further defined by Howard as a practice of increasing the efficiency of buildings and their sites, water, and materials and reducing building impacts on human health and the environment through better design and construction processes, maintenance, and removal [13]. Another significant definition is currently proposed in a new evaluation criterion in China, stressing on providing healthy environments, infrastructures, and public services to promote the health of humans and buildings (Healthy Building Evaluating Standard T/ASC02-2016) [13]. A healthy building environment is one in which an individual's health is neither jeopardized nor affected and one that promotes the health and well-being of the residents [14]. Healthy building environments can include residences, workplaces, or utility structures such as nursing homes, hospitals, and schools. Both indoor and outdoor environmental variables are addressed in healthy building settings to adapt effectively to occupants' health, requirements, and preferences. Adding value to the environment extends beyond sustainability for energy savings to a sustainable life for quality and excellent health. [15]. Table 1 presents the leading global leanings of healthy building definition and interlinked parameters.

**Table 1.** Leading global leanings of healthy building definition and interlinked parameters.

| Country | Years | Related Definition Paramaters | Data Source |
|---|---|---|---|
| Japan | 1980 | Healthy building is measured based on health as the benchmark and includes physical health and mental health as indispensable parts. | [16] |
| Japan | 1990 | An environmental symbiosis house gives full consideration to energy, resources, waste, etc. to protect the global environment and to create an intimate, beautiful and harmonious surrounding natural environment, enabling residents to live independently, healthily, and comfortably, while designing the house and its community environment. | [17] |
| / | 2000 | Healthy building is described as a kind of living environment embodied in the indoor living space and living environment, including not only physical environmental values, such as temperature, ventilation efficiency, noise, illuminance, air quality, etc., but also subjective psychological factors such as floor plan, space layout and color, privacy protection, landscaping, material selection, etc., plus job satisfaction, interpersonal relationships, etc. | [18] |
| USA | 2014 | Healthy building is described as committed to the pursuit of a built environment that supports human health and comfort, improving human health, mood, comfort, sleep, and other factors; encouraging a healthy and active lifestyle; and reducing the damage of chemicals and pollutants. | [19] |
| China | 2016 | Healthy buildings are buildings that provide people with a healthier environment and facilities and services based on satisfying building functions, promoting people's physical and mental health, and achieving improved health performance. | [20] |
| Europe | 2016 | Healthy building refers to fulfilling the basic requirements of the building; highlighting the health elements; meeting the occupants' physical, psychological, and multilevel social needs with the concept of sustainable development of human living health; and creating healthy, safe, comfortable, and environmentally friendly high-quality buildings and communities. | [21] |

Inasmuch as there is no unified definition for healthy building, on comparing the different definitions, it is worth noting that the key elements involved are about the same. The ideal healthy building addresses physical and psychological components and consists of key elements with high and low focus. Figure 1 summarizes the elements that contribute to healthy building. The key elements with high focus consist of indoor environment quality (IEQ), including thermal comfort; indoor air quality (IAQ); visual comfort; acoustic comfort; and ergonomics [18–20,22]. The key elements with low focus include water quality, material, services, security, community, and innovation [18–20,22], among others. Additionally, in order to sustain a healthy environment, supportive elements such as an energy conservation system and new technology must be considered, since their integration into the entire building system facilitates and enhances the building operation and maintenance throughout its lifecycle.

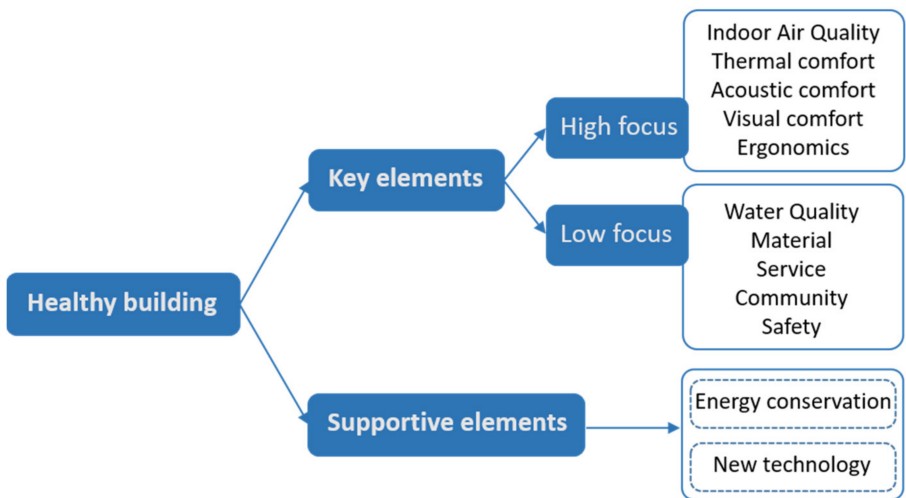

**Figure 1.** The key and supportive elements that contribute to healthy design.

Current Healthy Building Evaluation Systems Overview

While the concept of healthy building has been evolving over the last decade, the current implementation still remains at a preliminary stage [23]. In the evolvement process, healthy building standards have been separated into two categories: (i) design guidelines depicting a set of recommendations on how to apply design principles to provide a healthy environment and (ii) evaluation standards, which are methods of assessing the quality and performance of a building based on specific criteria (Figure 2). The design guidelines intervene during the construction process, while the evaluation standards are only required once the building is completed, during the operation stage. Based on evidence-based medical and scientific research, harnessing the built environment as a vehicle to support health and well-being has significantly contributed to making the WELL building standard the most prominent and widely applied globally [19,22,24]. Thus, the WELL building standard was given more prominence in our analysis, as it demonstrates the strongest potential for real-world application.

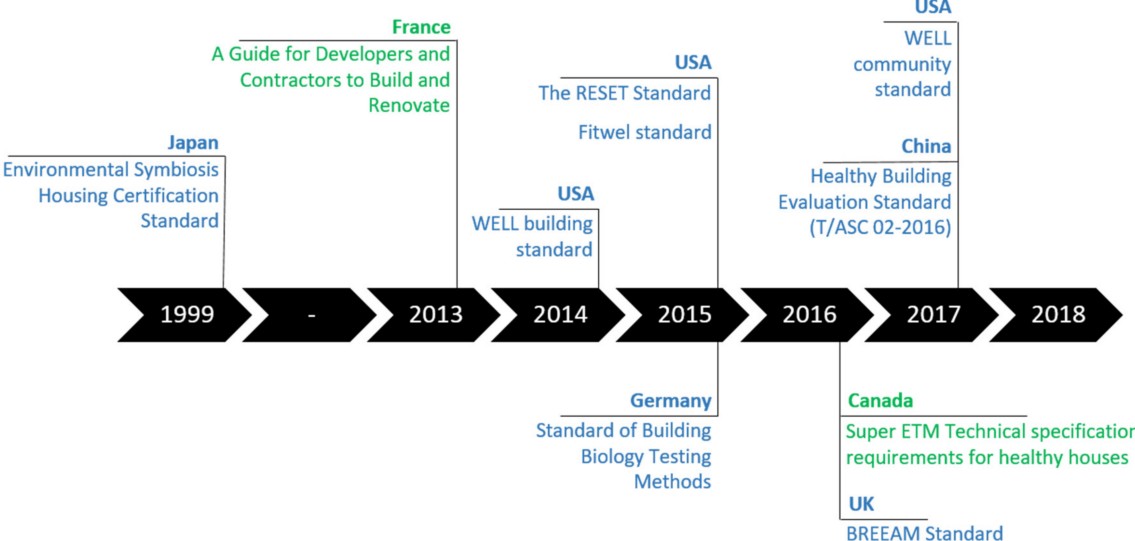

**Figure 2.** Evolution of healthy building design guidelines (in green) and evaluation standards (in blue).

## 2.2. Healthy Building Implementation in the Real World

Since the inception and launch of the healthy building concept in the 1980s [12], we examined the highlights of healthy construction implementation in a real-world project

cycle context (Table 2). It is evident that concepts related to healthy building are mostly applied during the construction phase of the project by addressing issues associated with indoor environment quality (IEQ) and/or indoor air quality (IAQ). The emphasis on the IEQ and IAQ components that entail air quality, thermal comfort, lighting, and acoustics, and how they interrelate, has constrained the full capacity actualization of healthy construction and accounted only for the physical aspect of healthy building. According to the above definitions, construction in the context of healthy building standards goes beyond IEQ and IAQ; it also accounts for more concepts like water, nourishment, movement, materials, innovation, and mind that guide human health related to the built environment, and it addresses both physical and psychological aspects [22,25–32]. Equally, the process implementation of healthy building standards from the design stage is rarely paid attention to, which shows the gap between the application of healthy strategy and the design process to achieve healthy building. This thus affirms that the standardized approach of IDP, though efficient, has fallen short of being aligned to the direct and robust achievement of healthy building concepts within IDP.

**Table 2.** Healthy building implementation and/or industry feedback in projects cycle framework from 1998 to 2021.

| Healthy Concept in Projects Cycle Framework and Key Findings | References |
|---|---|
| Healthy concept scope: Indoor air quality in healthy building. The indoor air quality of a building has a direct impact on the health and productivity of its occupants. There are various design solutions that may be implemented to provide optimal indoor air quality. However, it is critical to understand that, while indoor air quality is a crucial driver of "healthy design," it is not the only determinant. Other considerations include acoustics, vibration, light, comfort, aesthetics, and security, as well as ergonomic design features and safety. | [33–35] |
| Healthy concept scope: Indoor environmental quality in healthy building. Healthy building researchers have made significant advances in our understanding of the relationship between health and productivity. Indoor environmental quality is the primary requirement for occupant comfort and productivity in a structure. Thermal comfort, air temperature, humidity, radiation, internal illumination, air movement, activities, clothing, and climatic change are all components of indoor environmental quality that have a major impact on occupants. The quality of indoor environmental conditions may have significant economic impacts for our service society, which relies on buildings for workers to be productive. Additionally, it was found that there is a direct relationship between indoor air quality and building materials which contributes to the health of building occupants even though there are still many remaining uncertainties about the costs. | [25–27,36–39] |
| Healthy concept scope: Toward healthy building construction. There have been discussions around the integration of multi-disciplinary fields within indoor environmental science to achieve better indoor environmental quality. Building design decisions should be made with full awareness of the interdependent, dynamic interactions between indoor environmental factors and how they will be perceived by occupants in the ideal scenario. Stronger than ever, the indoor environmental specialist should make it his/her duty to get a seat at the project table from the beginning and throughout the whole design process, and to strongly protect the interests of the occupants. Furthermore, policy must be changed at several levels in order to promote healthy indoor settings. It is almost likely that the advantages of such initiatives, evaluated in terms of increased human health and productivity, much exceed the costs. | [40–43] |

On the other hand, it has already been proven that IDP is a holistic approach that integrates people, systems, business structures, and practices into a process that collaboratively harnesses the talents and insights of all participants to optimize project results, increase value to the owner, reduce waste, and maximize efficiency through all phases of design, fabrication, and construction [44–46]. Thus, informed by real-world experiences, in this study, we posit and advocate for a framework that aligns the entire design process (from Pre-Design to Post Occupancy) to achieve healthy building considering the gaps observed in the over 23 years of practice.

### 3. Aligning the IDP Continuum (from PD to PO) to Achieve Healthy Building

Although the IDP has so far been developed fully theoretically, having clear and general descriptions in addition to education on IDP in the contemporary university curricula being emphasized [47], its practical application is often far from smooth. There is no specific formula for the composition of an IDP team; depending on the nature of the project and the objectives to be accomplished, the required expertise may appear to be very diverse [47]. Moreover, the shortfall of IDP not having a robust health strategy (HS) capability is a concern of interest. Despite the well-intended aim of achieving high-performance building on well-defined environmental and social goals, the health impact of the built environment on health behaviors and disease transmission in social systems is gaining traction. Therefore, in the quantification of actors' input, there is an impending need to integrate a robust HS in IDP. Driven by these inherent and historical theoretical framework principles, a conceptual framework of IDP + HS aimed at invigorating the implementation of HS in all IDP workflow stages is advanced.

To achieve this framework, the first component examines the actual IDP concept to identify the various factors that come together to make up the entire process. It also shows the advantage of integrating any new system into IDP because of the collaborative network established among specialists right from the beginning of the design. In this research, the IDP is presented in 7 different phases (Pre-Design, Schematic Design, Design Development, Construction Documentation, Building Construction, Building Operation (startup), and Post Occupancy (long-term operation)) that portray the project delivery methods. The second part illustrates the integration of healthy building strategies into IDP by taking advantage of the feedback mechanisms occurring during the iterative process, which help evaluate and make appropriate decisions for the project to ensure healthy building is achieved. The iteration processes with feedback loops allow the design team and, mostly in this case, the healthy building specialist to oversee the whole process from pre-design (PD) to post-operation (PO).

*Integration of Healthy Building Strategy into IDP*

The process of advancing a refocused IDP for healthy building starts by examining the healthy building strategies and technologies during the construction phase. We developed a three-layer model that facilitates the process for healthy building construction activities based on the IDP project delivery method, even though the IDP project delivery method has not been standardized in implementing healthy building construction. To address these issues, additional elements are incorporated into the new IDP framework to highlight the areas where key team members could make a substantial contribution during the design phase by providing direct input to help achieve the goals of healthy building. The proposed IDP framework provides a comprehensive list of healthy construction activities, the goals and procedures at each design phase, and the roles and responsibilities of the team members. For this study, the Well Building Standard system requirements are used as a mean of assessment through design preconditions as it has been proven to be most influential healthy evaluation system with the widest application around the world [22].

The Well Building Institute (IWBI, New York, NY, USA) was created in 2013 with the mission to promote and improve the quality of human health and well-being within the environment. The Well Building Standards (WBS) are based on a medical study that examined the relationship between building and the health and wellness impacts on its occupants. It helps to create an environment that promotes good nutrition, exercise, mood, sleep, and the performance of the occupants [24]. The WBS was launched as a performance-based assessment system to assess, certify, and monitor buildings and communities that directly or indirectly impact human health. It provides quantifiable value to the health and well-being of the occupants living in the buildings. WBS outlines ten concepts applicable to the built environment, which are: air, water, nourishment, light, movement, sound, thermal comfort, materials, mind, and community [29].

To guarantee a healthy outcome, the collaborative process of IDP could be more focused on healthy construction by ensuring the integration of a healthy strategy through the entire process by a healthy building specialist. The need is supported by the fact that a healthy building specialist role is assigned to any potential partner building professional who has undergone extra training at the International WELL Building Institute (IWBI) as a WELL-accredited professional (WELL AP). The WELL AP credential denotes expertise in the WELL Building Standard (WELL) and a commitment to advancing human health and wellness in buildings and communities. The WELL AP credential is awarded to those who successfully pass the WELL AP exam [29]. This additional professional qualification justifies the requisite competence for the herein-suggested HS in the developed conceptual framework put forward by our research findings. Unlike the existing process of achieving healthy building by almost only including strategies for IEQ, the IDP + HS proposed in this study enables the healthy WELL-accredited professional to oversee the entire project component from PD to PO.

Figure 3 presents the overall iterative process of IDP and HS where the IDP + HS framework consists of three layers, which are the 'Design Layer as first one', the second being 'Construction layer', and the third as 'Operation Layer'. The process takes into account the Well Building Standard concepts as key elements which apply to the general building environment (outdoor and indoor) from PD to PO. New technologies are used to support the entire system with passive and active strategies implementation. The WELL AP, as the core member, plays a key role by leading the team and guiding them on how to achieve the expected results. The framework is organized by sets of simultaneous relationships.

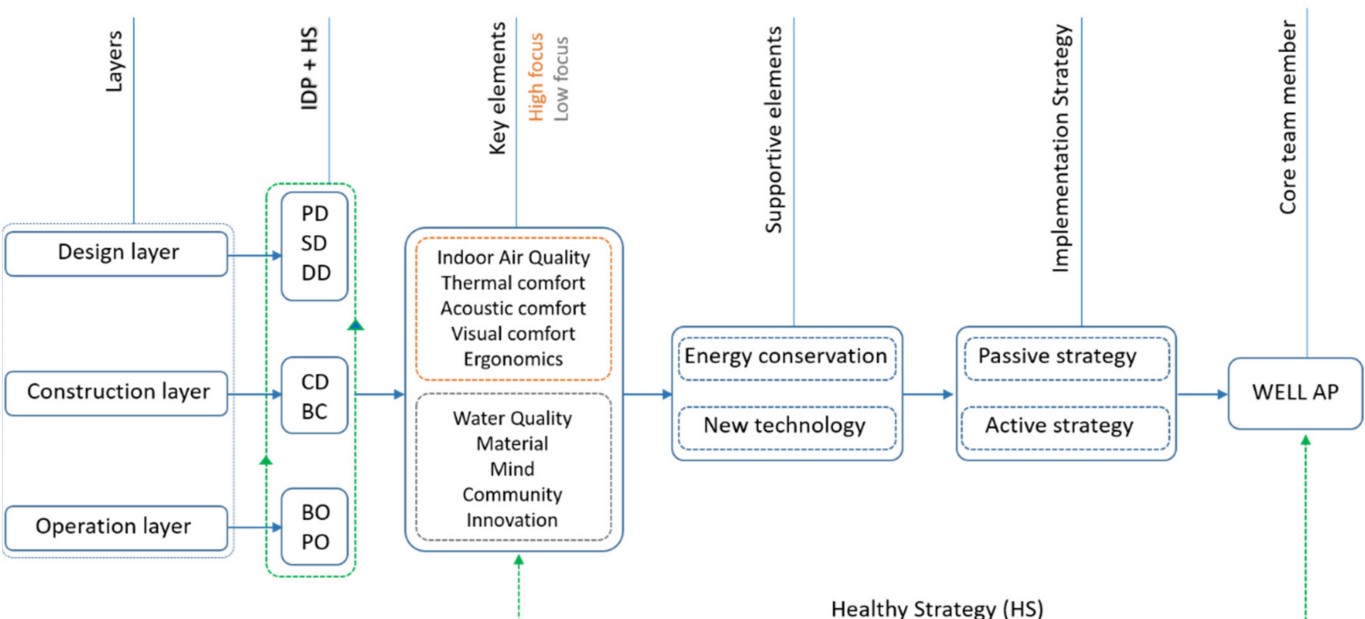

**Figure 3.** IDP layers process evolution and their interaction with healthy strategy.

The first layer of the IDP + HS framework (Figure 4), the 'Design Layer', focuses on the main achievement of the design team throughout the design process. This layer is divided into three phases: pre-design, schedule design, and design development. It pays maximum attention to the healthy building goals at this stage by providing a breakdown of the content of the built environment, including their precondition requirements, with the assessment methods needed to be implemented. At this stage, key elements are introduced to core team and tasks are assigned by the WELL AP, while constantly updating the project goal for each phase (PD, SD, and DD).

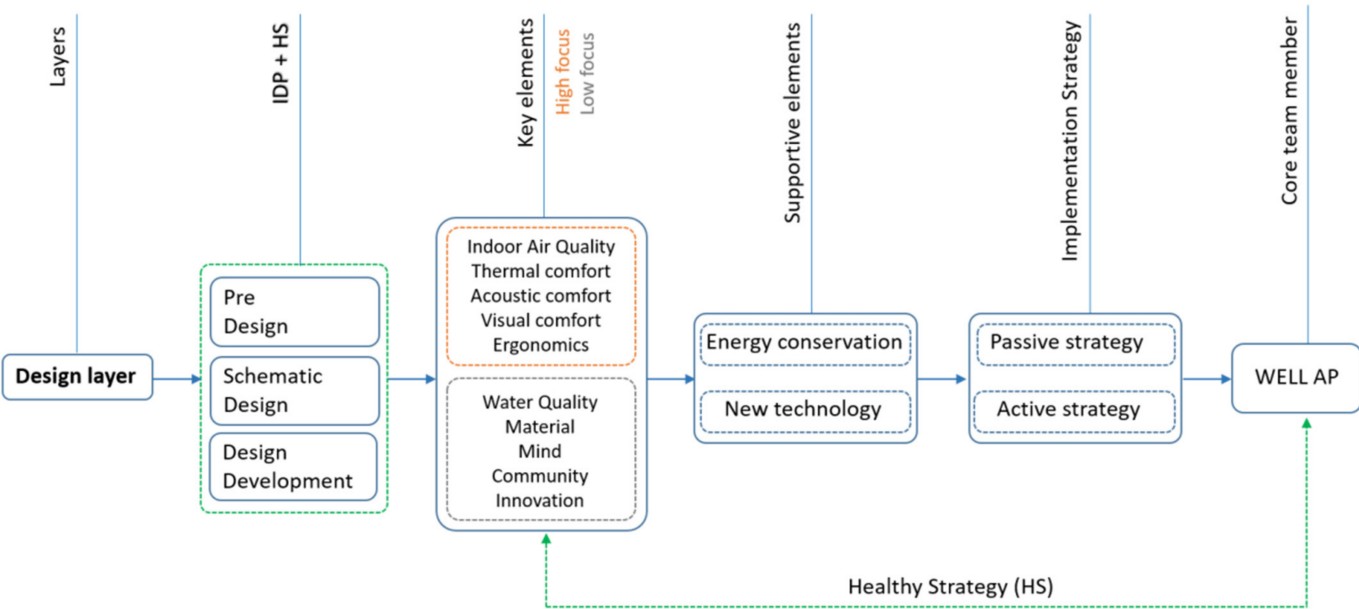

**Figure 4.** Design layer process evolution in IDP + HS.

The second layer of the IDP + HS framework (Figure 5), 'Construction Layer', focuses on the building construction. This layer is divided into two sections: Construction document and building construction. The main roles and responsibilities of the design team led by the WELL AP are to implement the intention of each healthy strategy applied during the previous layer (design layer). This phase deals with the implementation of healthy building strategy into the construction phase of the project and the application for the required document for healthy building certification. The WELL AP, in tight collaboration with the contractor and other specialists, leads the team toward healthy construction results by integrating each step within CD and BC.

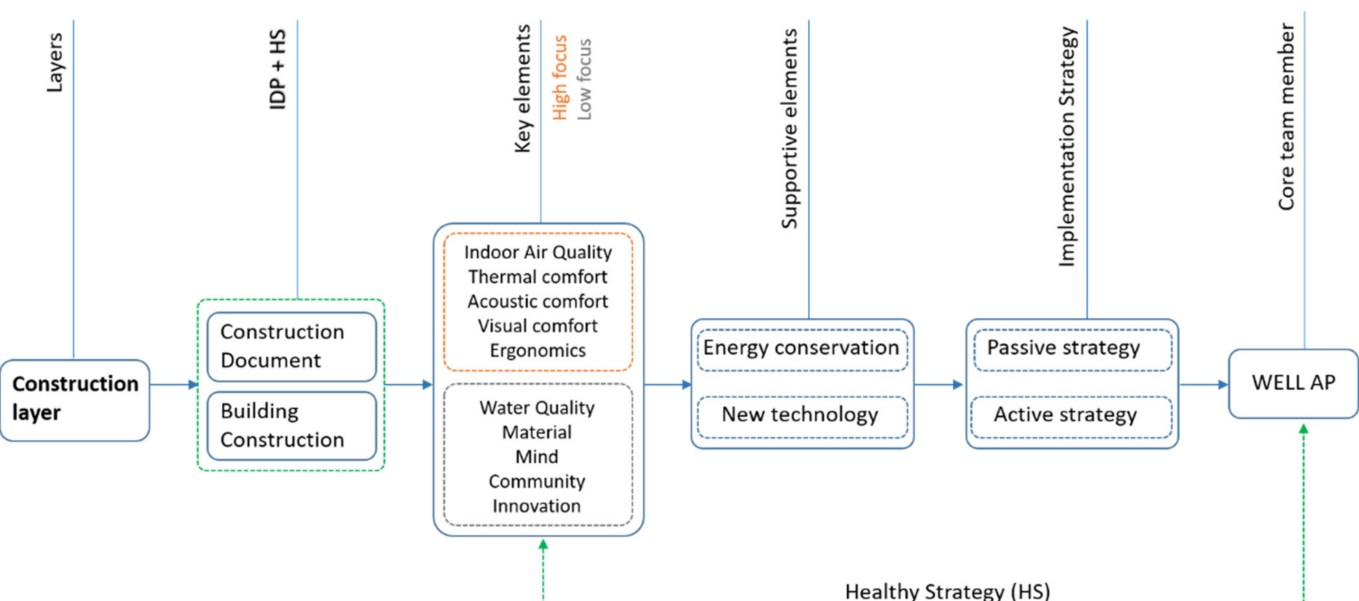

**Figure 5.** Construction layer process evolution in IDP + HS.

The third layer of the IDP + HS framework (Figure 6), Operation Layer, focuses on the building operation and post-occupancy. The main roles and responsibilities of the design team under the instruction of the health building specialist are to educate the building operators about healthy building operation and healthy behavior towards the facility.

Additionally, handoff and training sessions for building operators take place to facilitate the operation process, followed by a rigorous evaluation during the building operation and continuous assessment and maintenance during the post-occupancy.

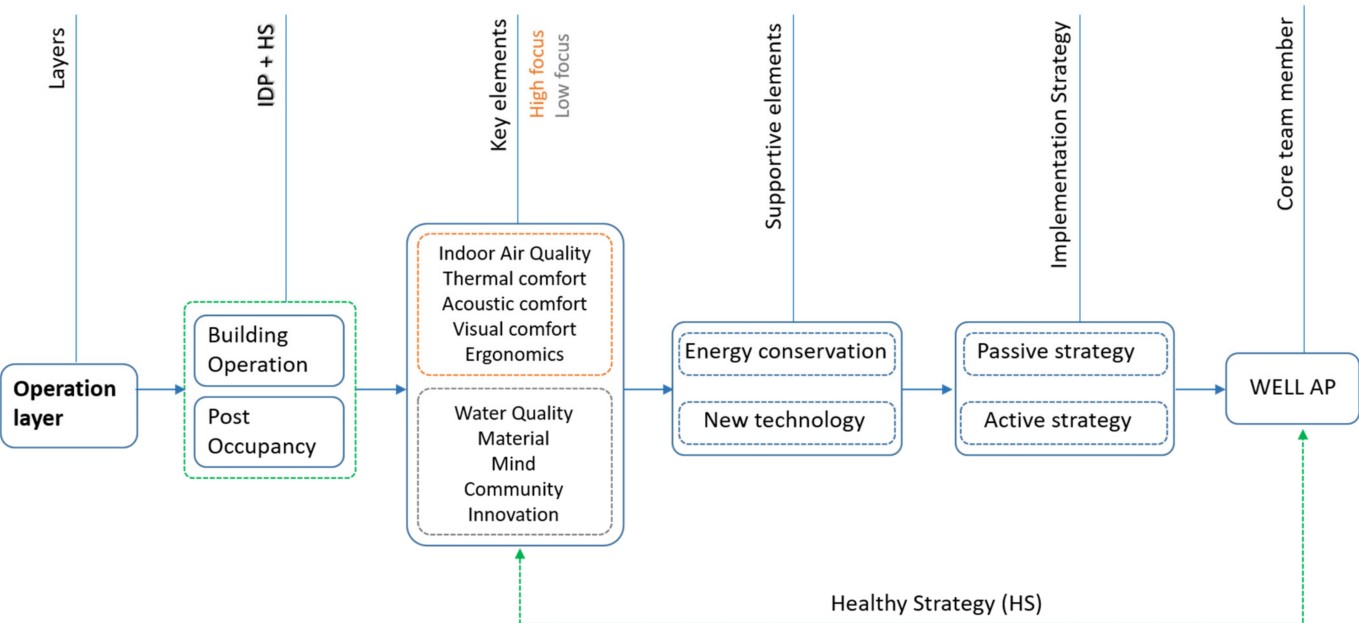

**Figure 6.** Operation layer process evolution in IDP + HS.

The process presents the entire IDP + HS framework. It illustrates typical IDP methodology as it relates to healthy strategy (HS) through an iterative feedback loop, allowing healthy building specialists to give maximum input from PD to PO. However, to permanently achieve these healthy goals, several facilitating policy adjustments or formulations would be desired.

## 4. Supportive Policies

Policies at the educational level must integrate IDP in the contemporary university and technical training curricula in the construction and urban planning field and must focus their teaching to appropriately prioritize IDP content with specific goals. To achieve healthy construction, for instance, teaching within IDP must focus on healthy strategy with the process. A practical and efficient way to facilitate IDP in the technical institute and university curricula is to create experimental learning opportunities for apprentices to learn from field experts. The pairing between trainees and professionals at the educational level can lead to easy integration into the professional network and enhance future collaboration on a specific construction project. Additionally, a program such as the WELL building standard, with its holistic approach to wellness in buildings that integrates healthy strategy into design, must be part of training curricula at all levels.

Policy makers at the international and national levels need to make the impact of the built environment on health a priority by promotion of healthy building where existing policies could be revised and updated accordingly while introducing cost-friendly incentives for healthy construction (for instance, tax exemptions and tax holidays) to allow the technology to thrive. The life cycle approach must be applied to the building industry management, where cooperation with local authorities must be prioritized with a clear focus on healthy building strategies and implementation, to ensure social and health fairness at all levels. Regulations governing building materials should be reviewed and updated where necessary to reflect the advantages of a healthy built environment while replacement of potentially hazardous materials and construction consumables with safer alternatives should be encouraged. Implementation of a healthy built environment procurement policy

based on agreed healthy building standards would help institute fairness for all at all levels of operation.

Policies in the building industry must lead to the application of healthy criteria in all the building systems where installation and management systems must be taken into consideration when applying healthy criteria. Rigorous networking and communication platforms for built environment end users, collaborating professionals, and experts, with information sharing on aspects of technological advancements, international and national policies, and strategic plan shifts, should be established and/or invigorated where systems exist to build a multi-directional communication platform with empowerment content in the sector for healthy living.

## 5. Perspective and Conclusions

Buildings have been proven to impact our health in different ways, with different factors. The concept of healthy buildings aims to improve the physical and mental health of people while saving energy and protecting the environment. Although buildings achieved through IDP can improve the health of occupants, their actions are still limited. Thus, the limits of today's building development must be addressed by integrating healthy building criteria, as the two concepts complement each other. Buildings developed through IDP fall in line with the green buildings concept and prioritize the consumption of resources and the holistic coexistence of man and nature, while healthy buildings emphasize the physical, physiological, and psychological health of the occupants, which is directly related to life quality. Additionally, the link between IEQ and IAQ and health, discussed in most studies, has been clearly shown by many researchers. However, they still lack a thorough approach, as they strictly focus on the physical aspect of design, at the construction stage and operation stage. Therefore, by addressing this gab, future research should use any baseline to develop a comprehensive and holistic framework that focuses on health and well-being from the design stage.

Moreover, the concept of healthy buildings becomes vital if we manage to achieve short and long-term goals by considering the indoor and outdoor environmental quality in the context of one's health. To successfully achieve healthy building practices, it is evident that a solid team must be composed of the project owner, architects, contractors, engineers, and a WELL AP who will benefit from a comprehensive iterative building framework that includes HS advances through the design, construction, and operation. As part of the effort to construct such a framework, this research posits further advancement of IDP in the context of incorporating HS in the prevailing IDP guidelines with a sequential iterative procedure, hence developing an IDP + HS workflow framework. Figure 7 illustrates the IDP + HS iterative framework, having an embedded WELL-accredited professional (WELL AP) shown in green with a feedback loop from PD to PO in IDP. The IDP + HS iterative framework herein advanced is intended to further aid neophyte and experienced building professionals to reflect about the process of achieving healthy building while optimizing IDP for one's health invigoration in the construction industry. Additionally, the IDP + HS framework proposed here will support the design team with the lead of the healthy building specialist to produce adequate construction documentation, an area that is often a problem for the construction project. The comprehensive documentation process for healthy construction included in IDP + HS will minimize the need for additional documentation specifically related to healthy building, greatly facilitating the certification process of healthy building in the construction industry. IDP + HS will thus facilitate and strengthen the relationship between IDP and health and the process, documentation, and implementation requirements to achieve a full healthy construction.

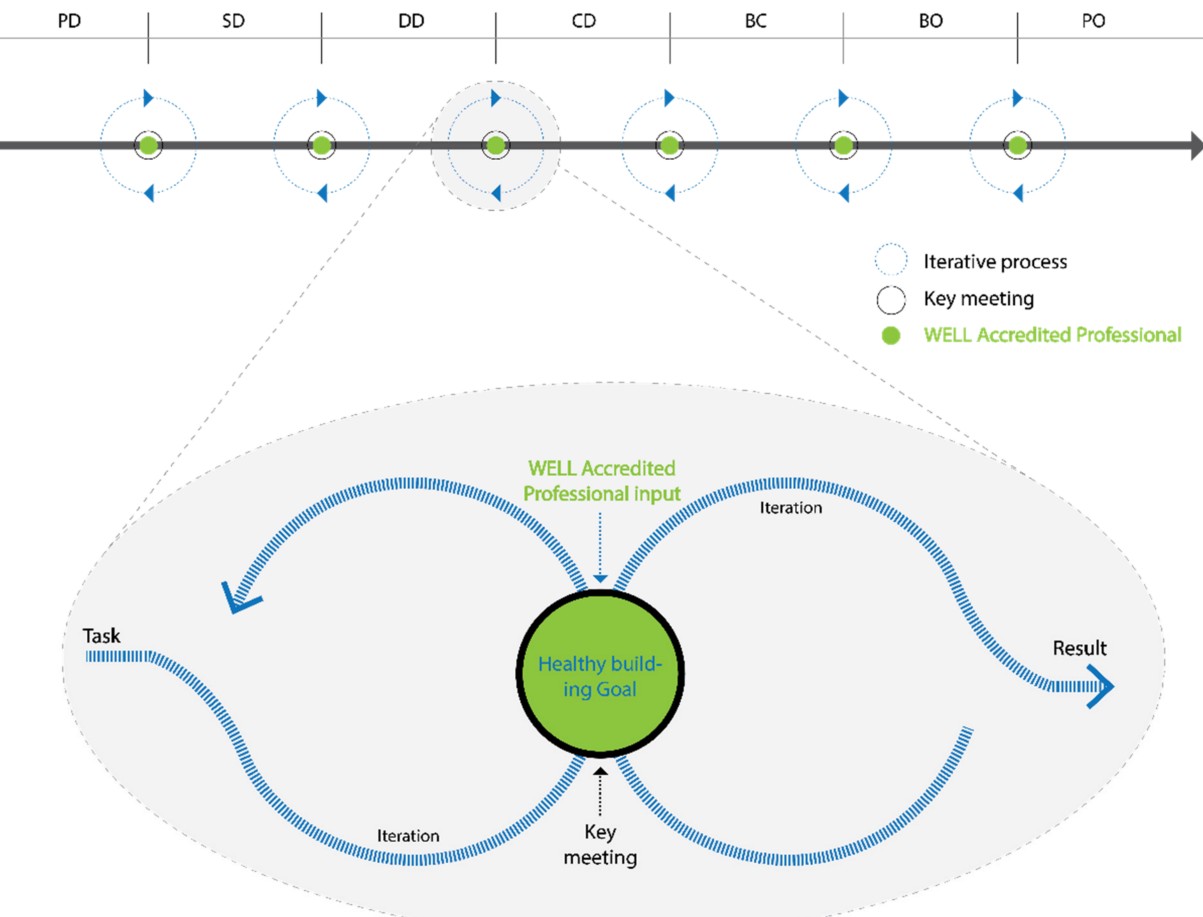

**Figure 7.** The IDP + HS iterative framework, having an embedded WELL-accredited professional (shown in green) with a feedback loop from PD to PO in IDP.

**Author Contributions:** Conceptualization: S.S.B.B.O.A. and L.Z.; Data mining and collation: S.S.B.B.O.A. and C.K.-W.; formal analysis: S.S.B.B.O.A. and C.K.-W.; writing: S.S.B.B.O.A., L.Z. and C.K.-W. All authors have read and agreed to the published version of the manuscript.

**Funding:** This research received no external funding.

**Institutional Review Board Statement:** Not applicable.

**Informed Consent Statement:** Not applicable.

**Data Availability Statement:** Not applicable.

**Acknowledgments:** The authors would like to thank all research team members for their valuable comments that help improve the paper.

**Conflicts of Interest:** The authors declare no conflict of interest.

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
