# Peer review of "Invigorating Health Strategy in an Integrated Design Process"

_smartcities, doi:10.3390/smartcities5030042_

Round 1

Reviewer 1 Report

The subject of the manuscript is important.

The manuscript is well presented and easy to understand.

However, the reviewer found no novelty in the proposed study. The review suggested that the author do a thorough literature review, clarifying the limitations of previous studies that these studies addressed.

It's important to qualify the manuscript when it is able to demonstrate its novelty and necessity.

Author Response

Response: Thank you for this observation. We included a review chapter which includes two sections. The first section is an overview of the healthy building concept definition at globally level, and the current status and evolution of healthy building evaluation standard and guidelines. The second section presents the healthy building implementation in real world, clarifying the current implication and the limitations of previous studies.

Again, we appreciate the opportunity to revise our work for consideration for publication in Smart Cities. We hope our revision meet your approval. We look forward to hearing from you in due time regarding our submission and to respond to any further questions and comments you may have.

Reviewer 2 Report

1_Originality and scope:

This article presents a relevant and necessary statement of principles for the scope of architecture and engineering. It is important to note that the health of the residents and users should be a top priority when designing, building and in terms of the following maintenance. Therefore, this paper is original and focuses on the new needs that society demands from the agents of the building process.

2_Literature and context:

The paper provides a promising start by which an international and multidisciplinary audience hopes to be provided with information on the environmental parameters that must be implemented in the design process, based on the scientific literature and in the current context that we are living globally. However, the study of this article does not obtain the information intuited at the beginning, the paper has not attempted to solve this issue either. The presented bibliographic references are not adequately integrated throughout the article. This results in providing a statement from a technician with the WELL rating as the only way to ensure the integration of the health strategy in the integrated design.

3_ Methods:

The method used in this paper is not suitable for achieving the proposed objectives. Indeed, this article’s purposes are very ambitious. The field of architecture does not need to critique the traditional design model; however, the field needs to be presented with new evidence and data to continue evolving. It is important to note that the traditional design method has never been linear, it has always been integrative. Architecture has always adapted to the changes demanded by society and has learned from the variety of experiences to improve the constructive process. At this present time, architecture must incorporate the factor of people’s health, thinking of users and keep collaborating hand in hand with engineers and contracting companies.

4_ Evidence / Results:

The conclusions of the article are not based on evidence because it does not provide data or reference dosimetry. Indeed, it simply shows that integrated design is better than traditional design and that the health strategy must be incorporated to reinforce integrated design.

5_Implications:

The implications of this article are aimed at university education, and I completely agree with the idea of integrating environmental parameters into the design process to improve people’s quality of life, since we spend 90% of our time residing inside buildings. Applying to either housing or workplace buildings.

6_ Communication:

The lack of communication in this article is evident and when one expects an aspect to become concrete, the content is blurred, and the direction is changed. A suitable article should be focused avoiding unnecessary distractions, for which we show these statements as proof:

Line 11_ Lack of guidelines in healthy buildings

There are numerous reference guides in healthy buildings and a summary with reference dosimetry was expected in each of the key elements described between lines 115 and 154.

Line 50_ Functional components of buildings from the envelope to furnishing have been thoroughly examined.

Line 71_IDP is a holistic and comprehensive

Line 88_The new IDP model serves useful education and training tool

Line 212_Lack of communication between architects and engineers.

Line 215-TDP implies sick building syndrome

7_ Conclusion:

This article focuses on two aspects: on the one hand on the organization and planning of the constructive process, and on the other on the need to implement the integrated design and reinforcing it with the health strategy. I agree with the two objectives, however, in order to achieve integration, contrasting data must be presented. This paper contains examples and reference standards that can be used. Therefore, I trust that the authors follow this magnificent path and show us the results of their research soon, since architecture has already changed without return and we need new methods.

Author Response

Comments and Suggestions for Authors

1_Originality and scope:

This article presents a relevant and necessary statement of principles for the scope of architecture and engineering. It is important to note that the health of the residents and users should be a top priority when designing, building and in terms of the following maintenance. Therefore, this paper is original and focuses on the new needs that society demands from the agents of the building process.

Response: Thank you for this observation.

2_Literature and context:

The paper provides a promising start by which an international and multidisciplinary audience hopes to be provided with information on the environmental parameters that must be implemented in the design process, based on the scientific literature and in the current context that we are living globally. However, the study of this article does not obtain the information intuited at the beginning, the paper has not attempted to solve this issue either. The presented bibliographic references are not adequately integrated throughout the article. This results in providing a statement from a technician with the WELL rating as the only way to ensure the integration of the health strategy in the integrated design.

Response: A section that presents the evolution and the current status of healthy building standard evaluation and guidelines has been incorporated.

3_ Methods:

The method used in this paper is not suitable for achieving the proposed objectives. Indeed, this article’s purposes are very ambitious. The field of architecture does not need to critique the traditional design model; however, the field needs to be presented with new evidence and data to continue evolving. It is important to note that the traditional design method has never been linear, it has always been integrative. Architecture has always adapted to the changes demanded by society and has learned from the variety of experiences to improve the constructive process. At this present time, architecture must incorporate the factor of people’s health, thinking of users and keep collaborating hand in hand with engineers and contracting companies.

Response: Thank you for pointing this out.

4_ Evidence / Results:

The conclusions of the article are not based on evidence because it does not provide data or reference dosimetry. Indeed, it simply shows that integrated design is better than traditional design and that the health strategy must be incorporated to reinforce integrated design.

Response: Thank you very much for this insightful comment.

5_Implications:

The implications of this article are aimed at university education, and I completely agree with the idea of integrating environmental parameters into the design process to improve people’s quality of life, since we spend 90% of our time residing inside buildings. Applying to either housing or workplace buildings.

Response: Thank you for this observation.

6_ Communication:

The lack of communication in this article is evident and when one expects an aspect to become concrete, the content is blurred, and the direction is changed. A suitable article should be focused avoiding unnecessary distractions, for which we show these statements as proof:

Line 11_ Lack of guidelines in healthy buildings

There are numerous reference guides in healthy buildings and a summary with reference dosimetry was expected in each of the key elements described between lines 115 and 154.

Line 50_ Functional components of buildings from the envelope to furnishing have been thoroughly examined.

Line 71_IDP is a holistic and comprehensive

Line 88_The new IDP model serves useful education and training tool

Line 212_Lack of communication between architects and engineers.

Line 215-TDP implies sick building syndrome

Response: Thank you for this observation, we have addressed that issue.

7_ Conclusion:

This article focuses on two aspects: on the one hand on the organization and planning of the constructive process, and on the other on the need to implement the integrated design and reinforcing it with the health strategy. I agree with the two objectives, however, in order to achieve integration, contrasting data must be presented. This paper contains examples and reference standards that can be used. Therefore, I trust that the authors follow this magnificent path and show us the results of their research soon, since architecture has already changed without return and we need new methods.

Response: Thank you for pointing this out, we have incorporated your suggestions throughout the manuscript.

Again, we appreciate the opportunity to revise our work for consideration for publication in Smart Cities. We hope our revision meet your approval. We look forward to hearing from you in due time regarding our submission and to respond to any further questions and comments you may have.

Round 2

Reviewer 1 Report

The reviewer agrees with the current form of the manuscript

Author Response

Comments

The reviewer agrees with the current form of the manuscript

Response: We appreciate your approval for consideration for publication in Smart Cities.

Reviewer 2 Report

Sorry, but I can see any special difference with the first version. Therefore, the paper needs new corrections.

Author Response

Comments and Suggestions for Authors

Sorry, but I can see any special difference with the first version. Therefore, the paper needs new corrections.

Response: We appreciate your observation; however the authors did not see any specific comments to rely on for the new corrections suggested, therefore some highlight were made on the previous comments
